# Differentiation between mechanically loose and fixed press-fit implants using quantitative acoustics and load self-referencing: A phantom study on shoulder prostheses in polyurethane foam

**Florian Vogl**[1]*, **Stefanie Greger**[1], **Philippe Favre**[2], **William R. Taylor**[1], **Paul Thistlethwaite**[2]

**1** Institute for Biomechanics, ETH Zürich, Zürich, Switzerland, **2** Zimmer Biomet, Winterthur, Switzerland

* florian.vogl@hest.ethz.ch

## Abstract

This study proposes to use cross-interface quantitative acoustics (ci-qA) and load self-referencing (LSR) to assess implant stability in a radiation-free, inexpensive, rapid, and quantitative manner. Eight bone analog specimens, made from polyurethane foam, were implanted with a cementless stemless shoulder implant—first in a fixed and later in a loose configuration—and measured using ci-qA under two load conditions. The loose implants exhibited higher micromotion and lower pull-out strength than their stable counterparts, with all values falling within the range of reported reference values. All acoustic characteristics differentiated between loose and fixed implants (maximum area-under-curve AUC = 1.0 for mean total signal energy, AUC = 1.0 for mean total signal energy ratio, AUC = 0.8 for harmonic ratio, and AUC = 0.92 for load self-referencing coefficient). While these results on bone substitute material will need to be confirmed on real bone specimen, ci-qA could ultimately facilitate the assessment of primary stability during implantation surgery and avoid unnecessary revision through quantitative evaluation of secondary stability during follow-up.

## Introduction

Total joint arthroplasty (TJA) is one of the most common orthopaedic surgeries, with over 650 000 total knee replacements [1] and 66 000 shoulder arthroplasties [2] performed per year in the United States alone. These numbers have steadily increased over recent decades [3]; a trend that is expected to continue considering the rising life expectancies and TJA use for younger patients [4]. To minimize the associated burden on the public health system and complications for the patients, it is crucial to develop methods that improve the surgical outcome.

In order that an implant can perform as part of the load bearing apparatus, the implant has to be fixed to the bone through a mechanically stable interface. This quality of the interface is determined by the cement for cemented implants, while non-cemented implants rely on

**Data Availability Statement:** All data and analysis scripts are available under DOI: https://doi.org/10. 3929/ethz-b-000353255.

**Funding:** Zimmer Biomet, Sulzerallee 8, Winterthur, Switzerland, (https://www. zimmerbiomet.ch/) provided support in the form of salaries for authors PF and PT and through providing experimental infrastructure but did not have any additional role in the study design, data collection and analysis, decision to publish, or preparation of the manuscript. The specific roles of these authors are articulated in the 'author contributions' section.

**Competing interests:** PF and PT are employees of Zimmer Biomet and are entitled to stock options. FV and WRT have applied for a patent regarding the "Non-invasive detection of implant loosening in total knee replacements using acoustic waves (T-2016-078)" with the European Patent Office. The authors declare no other competing interests. This does not alter our adherence to PLOS ONE policies on sharing data and materials.

osseointegration to bond the implant to the bone. Successful osseointegration requires sufficient primary stability of the implant, whereas insufficient primary stability, considered to be related to excessive micromotion between implant and bone, leads to the creation of interfacial fibrous or fibrocallus tissue, and reduces the stiffness of the interface [5,6]. Despite the known importance of primary stability, methods for its intra-operative assessment remain sparse, often relying solely on the surgeon's experience. Additionally, even implants that have achieved primary stability can loosen over time and require another surgery, which accounts for 40% to 80% of revisions [7,8]. Here, it is often difficult for the clinician to differentiate between a loose implant and other factors that may cause pain but are not related to loosening (e.g. overloading of the ligaments or surrounding soft-tissues). As a result, pain in the implant region may be unnecessarily treated through revision surgery instead of applying a more conservative treatment. It is therefore clear that the assessment of primary stability during surgery and an evaluation of secondary stability at follow-up could both profit from a quantitative approach to assess the integrity of the bone-implant interface.

Traditionally, implant loosening is detected using x-rays when radiolucent lines appear at the implant interface [9]. However, with detection rates of only 20% for a radiolucency thickness of 0.3mm [10], this technique cannot reliably detect early implant debonding. While other approaches, such as subtraction arthrography [11], radionuclide arthrography [12–14], single photon emission computerized tomography associated with CT-scan (SPECT) [15], $^{18}$F-Fluoro-deoxyglucose positron emission tomography, and bone scintigraphy [16,17], have shown sensitivities and specificities between 50% and 90%, aseptic loosening remains particularly hard to detect [16]. Furthermore, these techniques either expose the patient to ionizing radiation, require the injection of additional agents, take longer than 30 mins, suffer from considerable inter-observer variability [11,18], or are expensive, and thus have limited applicability for inter-operative assessment.

To address these issues, acoustic techniques have been proposed as alternative means to directly assess the mechanical properties of the bone-implant interface. Based exclusively on the mechanical phenomenon of acoustic wave propagation, acoustic techniques are radiation-free, non-invasive, and can offer a quantitative and rapid assessment. Here, different acoustic approaches based on harmonic generation, coherence-based measures, spectral amplitude, resonance shifts, peak flattening, and peak counting have shown promise for detecting implant loosening [19–23]. One of the key challenges in bringing these techniques into clinical practice is their dependency on the target joint, implant type, and individual anatomy, which makes the identification of universal measures and absolute criteria for assessing implant stability difficult.

We therefore propose a new approach to address these issues: load self-referencing (LSR). LSR is based on the hypothesis that application of a load to the implant changes the acoustic properties of an unstable interface (e.g. through closing small gaps between the implant and bone), while not affecting a stable interface. Thus, measurement of the implant under different loading conditions would allow the acoustic characteristics to be evaluated with respect to changes arising from loading conditions, rather than in absolute terms. In this approach, the individual patient and implant (target joint, implant design. . .) serve as their own reference and are thus automatically taken into account, rendering comparisons against a population distribution unnecessary. LSR could easily be translated into clinical practice through e.g. measurement with different loads applied by the surgeon during implantation or, for assessing secondary stability, measurements with the patient in a loaded (e.g. single-legged stance for hip and knee implant assessment) versus a relaxed (e.g. sitting) position at follow-up. While LSR can be applied to nearly every measurement procedure, we secondarily propose to measure the acoustic wave propagation across a specific bone-implant interface by positioning sensors

on both the implant and the bone. Compared to other setups in which sensors are only placed on either the bone or the implant, this configuration allows to measure both transmissions and reflections through a specific bone-implant interface. As a result, cross-interface measurements could improve loosening detection and ranging.

As a first baseline evaluation of these proposed approaches, the goal of the presented study was to investigate two hypotheses: a) cross-interface acoustic measurements can differentiate between loose and fixed press-fit implants, and b) LSR is able to improve differentiation performance over metrics derived from single-load acoustic measurements.

## Materials and methods

### Overview

For this study, 8 bone analog specimens manufactured from polyurethane foam were implanted with a cementless, stemless shoulder implant; firstly in a fixed and later in a loosened configuration, as quantified through micromotion and pull-out strength tests, which represent the current gold standard techniques for the mechanical laboratory assessment of primary stability. All specimens were measured across the bone-implant interface using quantitative acoustics under two load conditions (no loading/physiologically-based loading), allowing for load self-referencing. Measurement uncertainties were estimated through independent repeatability measurements with full repositioning of all parts of the measurement setup on a single test specimen.

Sidus® stem-free shoulder implants (Zimmer Biomet®, Winterthur, Switzerland), consisting of a titanium alloy humeral anchor and a cobalt chromium alloy humeral head, were used in this study.

We have chosen an uncemented stemless implant [24,25] because in these implants success depends critically on whether primary stability can be achieved (as compared to e.g. cemented implants, where the cement provides additional support). Therefore we considered such an implant as a challenging, but also relevant case in which the presented method might be applied in the future.

Even though these simplified systems can clearly only be considered an approximation to the clinical case, foam bone substitutes are commonly applied to investigate implant-related questions [19,26,27], as long as key mechanical characteristics are in good agreement with clinical values. For our study, the foam bone's density of 0.32 g/cm$^3$ is similar to the bone density found in the healthy proximal humerus of 0.24–0.3 g/cm$^3$ [28] and satisfies the bone requirements of the Sidus® stem-free shoulder implants [29]. Furthermore, our specimen's micromotion and pull-out-strength values fell conservatively within the range of reported clinical values (see Results and Discussion). Lastly, the use of foam bone allowed for tightly controlled bone material properties and avoided the effect of variable bone quality.

### Specimen preparation

The bone analog specimens (Sawbones, Malmö, Sweden) consisted of polyurethane foam in a cylindrical shape with a radius of 55 mm and a height of 40 mm. A cavity, geometrically identical to the tool used to create the cavity surgically, was machined into each specimen for implantation of the anchor (Fig 1, left). For experimental handling, the bone specimens were then fixed into a PMMA bone cement cylinder made from osteobond copolymer bone cement (Zimmer Inc., Warsaw, Poland) in a two-stage process, taking special care to control the specimens' relative position and orientation, avoid any air enclosures, and keep the upper surface free from cement. This bone cement was chosen to minimize the acoustic impedance mismatch between cement and foam bone, with only 4% of energy being reflected from this

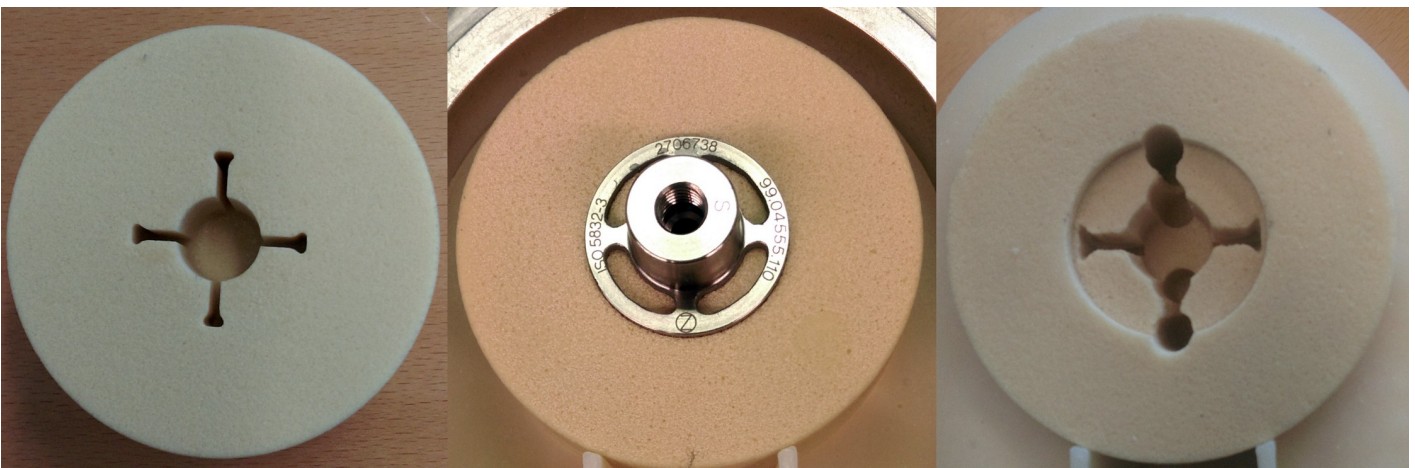

**Fig 1. Top view of a bone analog specimen and the machined cavity for the "fixed" condition (left), with implant anchor inserted (middle), and after removal of support material for the "loose" condition.**

interface. Access to the bone specimen for sensor mounting was enabled through a slot in the cement (Fig 1, right).

An axial loading machine (Z020, Zwick/Roell, Ulm-Einsingen, Germany) pressed the implant anchor into the pre-milled cavity with a maximum force of 5300 N and a maximum displacement rate of 10 mm/min, creating the "fixed" press-fit implant condition (Fig 1, middle). Using these press-fit specimens, acoustic measurements, micromotion, and pull-out tests were performed (see below). To create the "loose" implant condition, a countersink was used to remove a ring of foam bone support from under the implant anchor ring (Fig 1, right). Additionally, four bore holes of 5 mm diameter and 18 mm depth were drilled below the two anchor fins lying in the loading plane of micromotion tests. Finally, the implant anchor was reinserted using the same procedure as in the fixed case, using a maximum force of 750 N.

## Micromotion and pull-out tests

For the micromotion tests, the specimens were mounted at a 30˚ angle in the sagittal plane (Fig 2) and loaded cyclically between 50 N and 870 N at 1 Hz for 300 cycles using a hydraulic MTS 858 Table Top system (MTS Systems, Eden Prairie, MN, USA), following a previously published method [24].

Four differential variable reluctance transducers (DVRT, SG-DVRT-8, LordMicrostrain, Cary, NC, USA; 2μm resolution) were clamped around the implant head and measured the relative displacement between implant and bone surface throughout the load cycles assuming rigid-body kinematics. Data was acquired at 600 Hz using Spider 8–30 TF measurement electronics running CatmanEasy 3.4.2.52 software (both HBM, Darmstadt, Germany). Micromotion was calculated as the arithmetic mean displacement over the first 20 load cycles to ensure comparability between the micromotion and the preceding acoustic measurement.

For the pull-out tests, the implant head was removed and a uniaxial loading machine (Z020, Zwick/Roell, Ulm-Einsingen, Germany) pulled out the implant anchor from the foam bone at a displacement rate of 10 mm/min. The pull-out force was defined as the maximum force measured during this procedure.

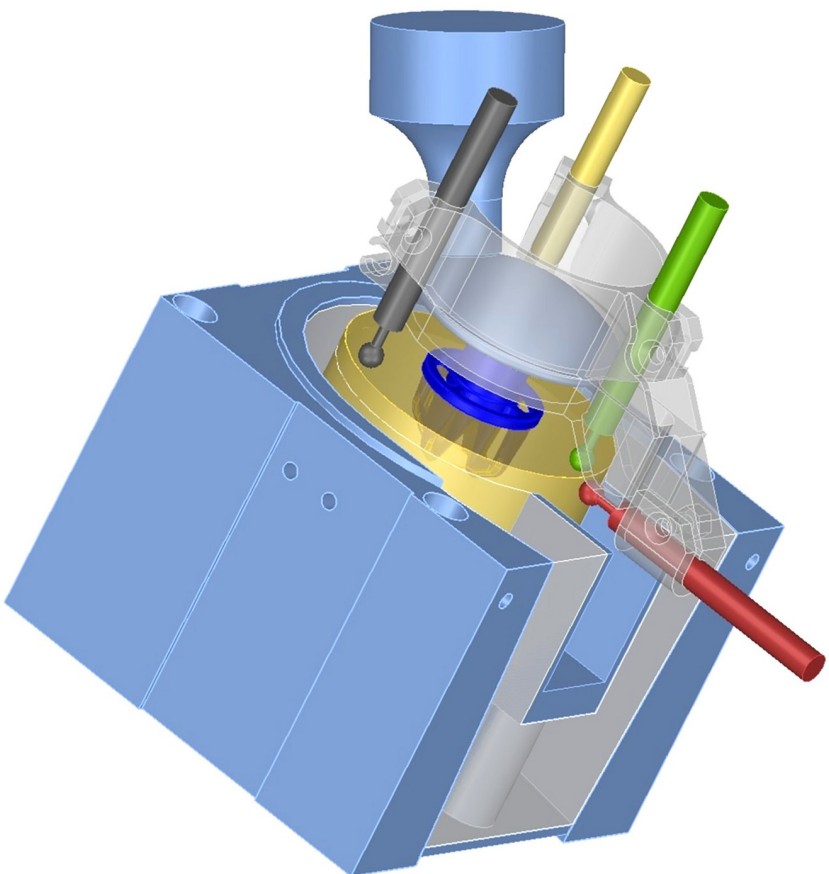

**Fig 2. Setup for micromotion testing; bone specimen (gold), implant anchor and head (dark blue), the four differential variable reluctance transducers (red, green, yellow, dark grey), bone cement (light grey); mounting and loading device(light blue).**

## Quantitative acoustic measurements

The device to perform the quantitative acoustic measurements was an adaptation of an established device used for axial transmission quantitative acoustics on human long bones [30,31]. This device consists of three main parts: a piezo-electric actuator to create the wave, four accelerometers to measure the wave propagation, and a portable electronic unit to handle data acquisition and device control.

The actuator consisted of a P-840.20 piezo-stack (PI Ceramic GmbH, Lederhose, Germany) with a custom-made stainless-steel pin head, which were enclosed in a custom-made housing. A spring mechanism was used to ensure that the contact pressure between actuator and implant remained at 30 N ± 1 N for all measurements. A sine wave with central frequency of 3.5 kHz, enveloped by a Gaussian of Full-Width-at-Half-Maximum of 2.5 kHz, drove the actuator after amplification by an E-617 high-power piezo-amplifier (PI GmbH & Co. KG, Karlsruhe, Germany). The wave propagation was measured using four 4516 accelerometers (Brüel & Kjael GmbH, Pöcking, Germany), placed at predetermined positions using custom-made guides for reproducibility (Fig 3): on the taper of the implant opposite the actuator ("implant sensor"), on the bone resection plane ("bone sensor"), and at two different heights on the side of the cylindrical foam bone specimen ("inferior and superior side sensors").

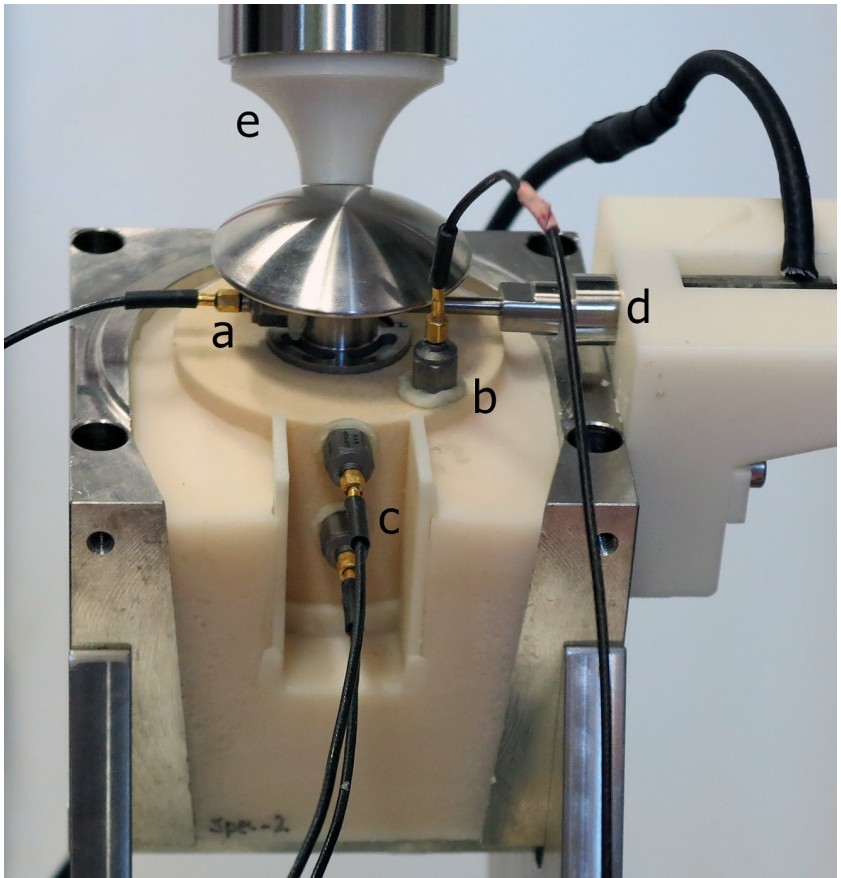

**Fig 3. Setup for the quantitative acoustic measurements; a) implant sensor, b) bone sensor, c) superior and inferior side sensors, d) transducer and transducer head, e) loading device.**

The signal generation for the actuator and the data acquisition from the accelerometers were handled by a USB-4431 data acquisition card (National Instruments) with a sampling rate of 96 kHz and an acquisition length of 2000 data points, corresponding to 20.8 ms. The device was controlled by a laptop running in-house software, programmed in Labview 13 (National Instruments, Austin, Texas), which handled the signal generation and storage of raw measurement files for later analysis. All raw measurements and figures of the corresponding spectra can be found in the electronic supplementary material.

For each of the quantitative acoustic (QA) measurements, the bone specimens were mounted at a 30˚ angle in the sagittal plane (Figs 2 and 3) and loaded at 50 N for the "unloaded" condition and at 870 N for the "loaded" condition, corresponding to physiological loading values found in the shoulder during various activities of daily living [32]. The following acoustic characteristics were computed for each measurement:

- Mean total signal energy: Arithmetic mean of the squared acceleration signal of each sensor over the total measurement time. Please note that this quantity does not directly correspond to the energy of the propagating waves.

- Mean total signal energy ratio: Ratio between the mean signal energy in each bone sensor and the implant sensor, which corresponds roughly to the fraction of signal energy transmitted from the implant to the bone.

- Harmonic ratios: Ratio between the first and the second harmonic were calculated for all sensors as

$$R_{0,1}^{3500} = \frac{\sum_{f=2.5kHz}^{4.5kHz} A(f)^2}{\sum_{g=6kHz}^{8kHz} A(g)^2} \tag{1}$$

where $A(x)$ is the value of the spectral component at frequency $x$, as determined by a Fast-Fourier-Transform with 16000 points, corresponding to approximately 10 times the signal length.

For the mean total signal energy and mean total signal energy ratio, a load self-referencing coefficient was calculated as:

$$LSR = \frac{X_{loaded} - X_{unloaded}}{X_{loaded} + X_{unloaded}} \tag{2}$$

with X being the acoustic characteristic (mean total signal energy, mean total signal energy ratio, or harmonic ratio).

## Estimation of measurement uncertainties

As previous studies have clearly shown that repositioning of specimen, actuator, and sensors is a main source of error in QA-measurements [30], measurement uncertainties arising from the repositioning of the multiple specimen and implant conditions were estimated as follows: The test specimen was measured eight times with full repositioning of each part of the experimental setup and the coefficient of variation (CV) was estimated for each acoustic characteristic. As these CVs surpassed the variations obtained from multiple measurements without full repositioning by orders of magnitude, the CVs were assigned as measurement uncertainties to each single measurement during the main experiment.

Uncertainties for the load self-referencing coefficients were calculated based on Gaussian error propagation as

$$\Delta LSR = 2 \sqrt{\frac{X_{unloaded}^2 \Delta X_{loaded}^2 + \Delta X_{unloaded}^2 X_{loaded}^2}{(X_{loaded} + X_{unloaded})^4}} \tag{3}$$

With $\Delta X$ being the estimated measurement uncertainty for acoustic characteristic $X$ under loading $i$.

## Statistics

Paired t-tests were used to compare each acoustic characteristic between the fixed and loose implant conditions for each sensor and loading state. The p-values were Bonferroni-corrected based on the total number of t-tests performed in our study (n = 35). This approach leads to a very conservative estimate of the p-value, as many of these tests are not independent from each other. Corrected p-values smaller than 0.05 were considered statistically significant. Standard Receiver Operating Characteristics (ROC) and area-under-curve (AUC) values were used to quantify whether acoustic characteristics could differentiate between fixed and loose implant states.

## Results

The loose implants showed higher micromotion and lower pull-out-strength than the stable implants: Micromotion was 67.7 μm ± 2.6 μm (mean ± standard deviation) for the fixed and

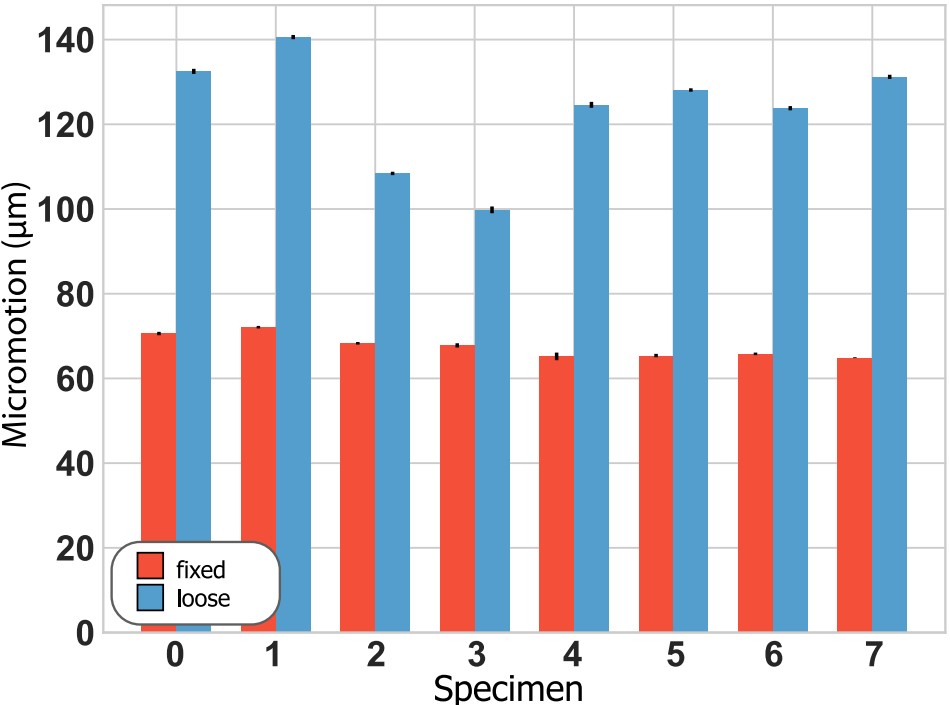

**Fig 4. Average micromotion for each specimen.** Error bars depict the standard deviation over all load cycles.

123.7 μm ± 13.3 μm for the loose implant condition (Fig 4), while pull-out strength was 758 ± 76 N and 271 N ±+/- 60 N (Fig 5), respectively.

All three acoustic characteristics differentiated between loose and fixed implants, with differentiation performance varying between sensors. Mean total signal energy achieved a maximum AUC = 1.0 (Fig 6), mean total signal energy ratio an maximum AUC = 1.0 (Fig 7), and harmonic ratio a maximum AUC = 0.8. Only the load self-referencing coefficient based on the mean total signal energy in the implant sensor achieved statistical significance and differentiated loose and fixed implants with an AUC of 0.92 (Fig 8). Table 1 presents the differentiation performance of each acoustic characteristic, sensor and loading.

## Discussion

The presented study investigated whether a) cross-interface acoustic measurements (ci-qA) can differentiate between loose and fixed press-fit implants, and whether b) load self-referencing (LSR) improves performance over using metrics derived from single-load acoustic measurements alone. Our results showed that all acoustic characteristics measured by ci-QA differentiated between loose and fixed implants. Although LSR differentiated between implant states, the benefits of combining load self-referencing with other acoustic characteristics derived from single-load measurements could not be confirmed and remains to be further investigated.

Our results demonstrate that the mean total signal energy (MTSE) (Fig 6) in all sensors, as well as the mean total signal energy ratio (MTSER) between implant and bone sensor (Fig 7), were able to differentiate between loose and fixed implant states irrespective of loading (Table 1). The acoustic characteristics appear consistent in pattern across all sensors, even

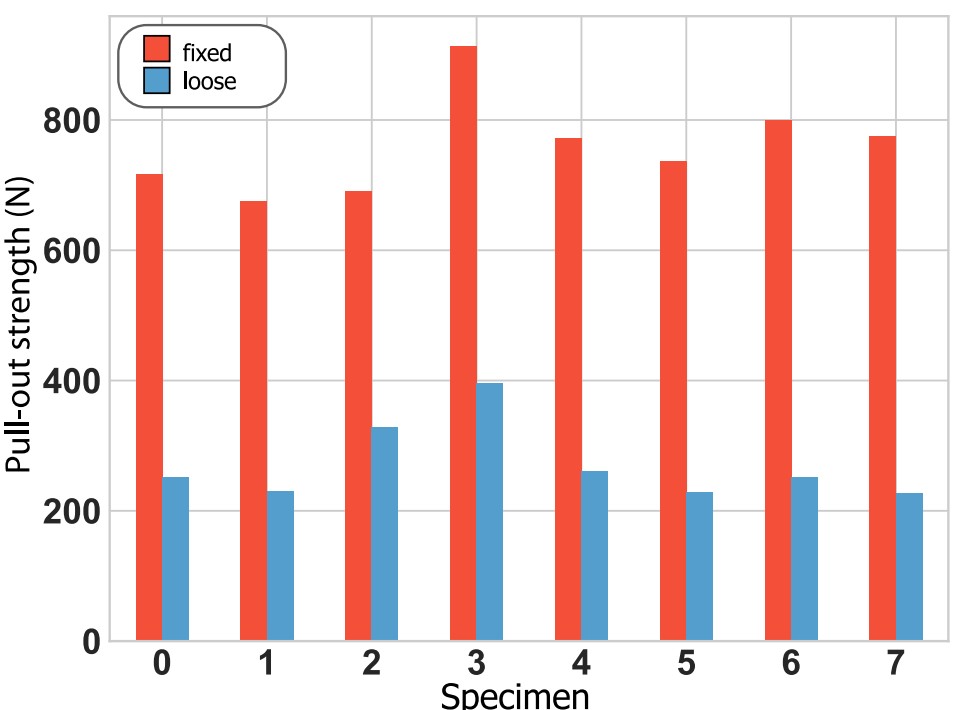

**Fig 5. Pull-out strength for each specimen.**

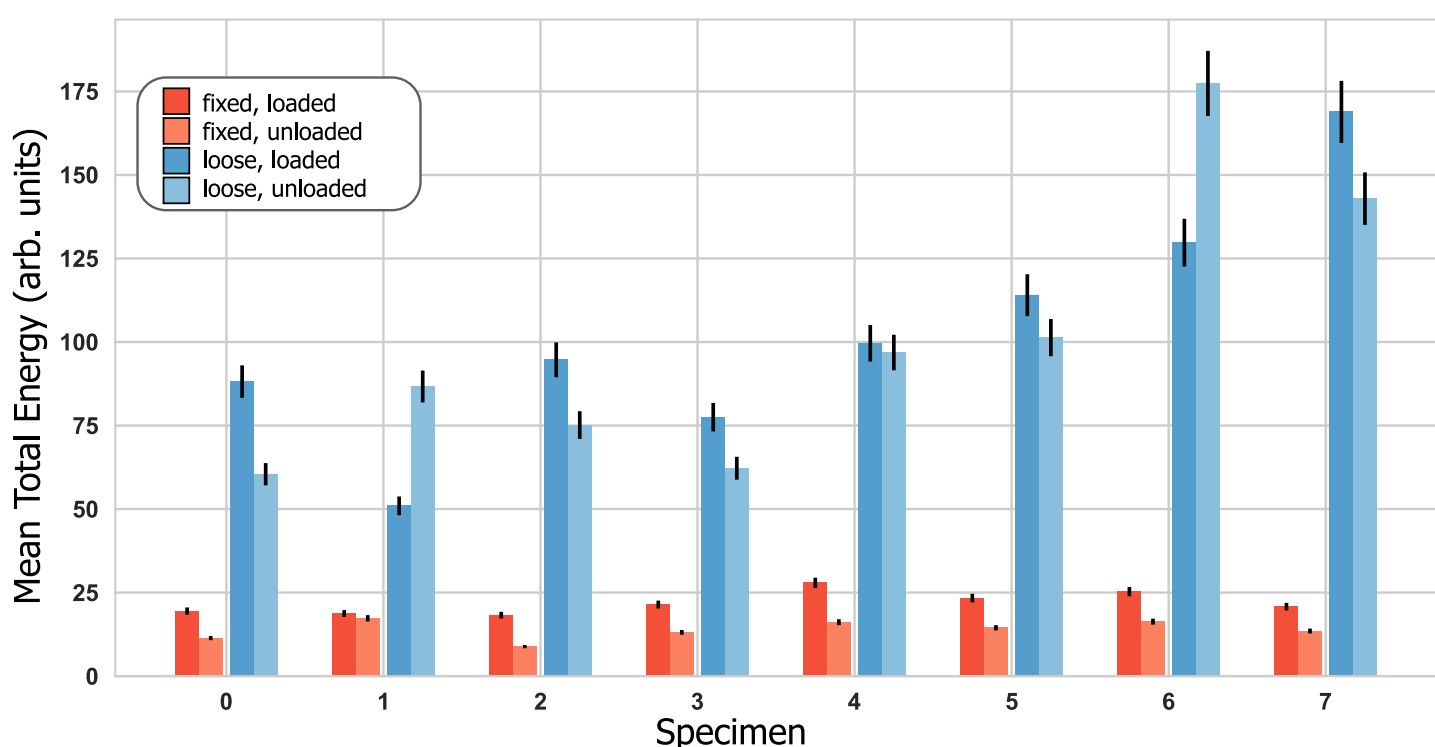

**Fig 6. Mean total signal energy for the implant sensor with the error bars depicting the estimated uncertainty.** The corresponding ROC-AUC values were 1.0 (loaded) and 1.0 (unloaded).

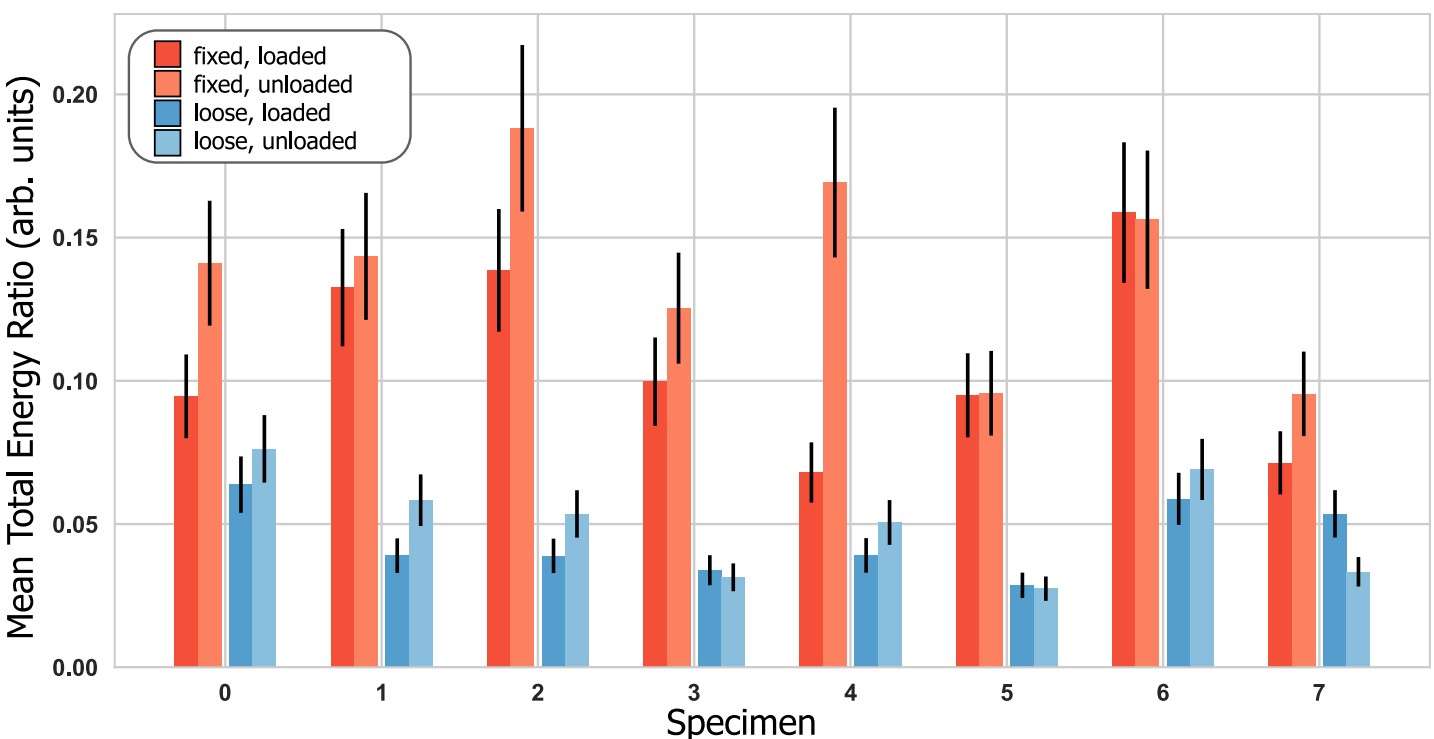

**Fig 7. Mean total signal energy ratio between the bone sensor and the implant sensor with error bars depicting the estimated uncertainty.** The ROC-AUC values were 1.0 (loaded) and 1.0 (unloaded).

though the differentiation power was reduced for the side sensors (Table 1). Deviations between sensors may have been attributable to differences in directionality and placement of the sensors, which could lead to different patterns of interference for the reflections propagating inside the bone, as well as different sensitivities to measure specific wave modes. The measured MTSE-values and their consistency across sensors indicate that an acoustic measurement imparts about 3 to 5 times more signal energy into a loose than into a fixed implant system (Fig 6). From this signal energy, a loose implant-bone interface transmits around 3–6% to the bone while a fixed interface transmits more than twice that amount (Fig 7). It therefore seems that the interface loosening changes the bone-implant system's mechanical properties, possibly reducing the structural stiffness and apparent mass [33], and allows more signal energy to be passed from the transducer to the implant when the implant is loose. However, even though more overall signal energy is imparted to the system for a loose implant, the strong mechanical coupling of a fixed implant transmits a higher fraction of this signal energy from the implant to the bone. In other words, loosening seems to increase the signal energy transferred from the transducer to the implant but decreases the signal energy transmitted across the bone-implant interface.

These findings agree with another sawbone study on hip implants, which found that replacing the femoral stem with a smaller, slightly lighter stem led to an increase in signal energy at the trochanter when the transducer was placed at the femoral condyle, and to a decrease at the iliac crest [19]. Interestingly, acetabular cup loosening, created through abrading the exterior side of the cup, could not be detected. Here, our results suggest that the signal changes truly reflect the reduced coupling from the smaller implant rather than the mass decrease. Based on

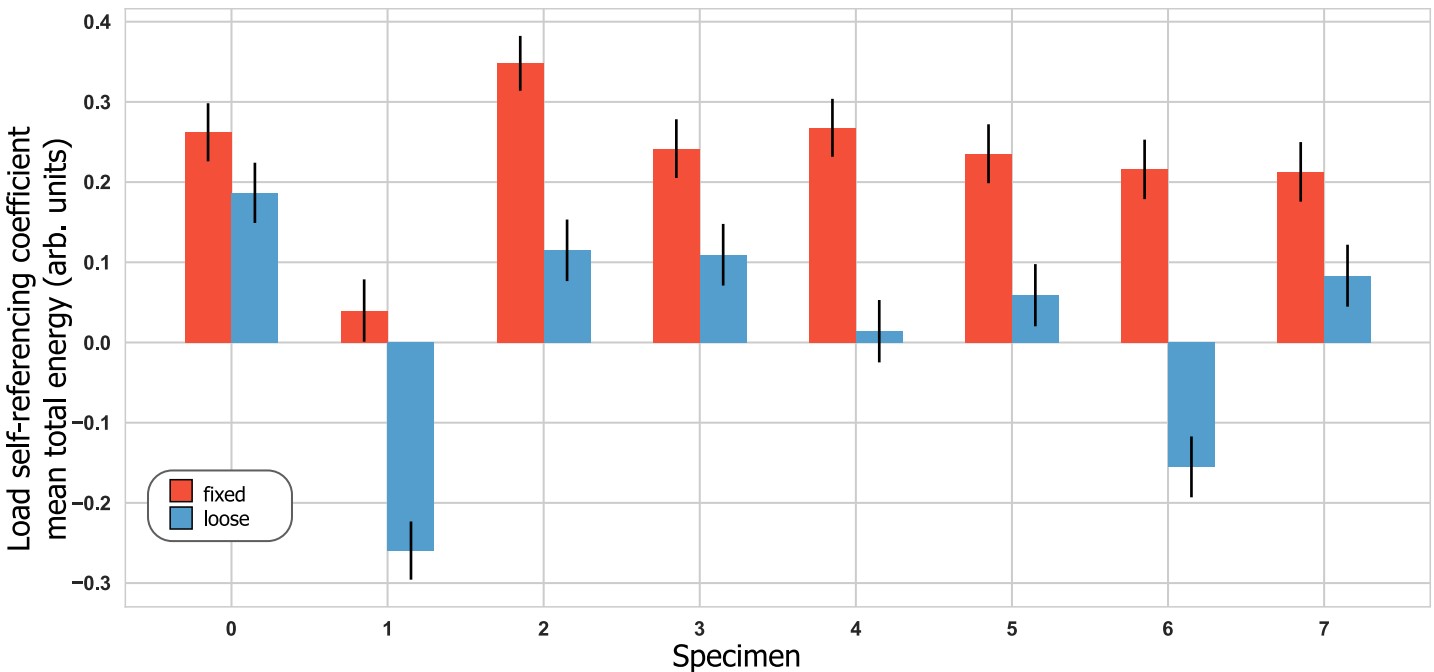

**Fig 8. Load self-referencing coefficient for the mean total signal energy in the implant sensor with error bars depicting the estimated uncertainty.** The corresponding ROC-AUC was 0.92.

our results, we furthermore speculate that abrading the exterior side of the acetabular cup did not yield a sufficiently loose state (e.g. in terms of micromotion) to be detected and that higher levels of loosening would further decrease the signal energy measured at the iliac. Under this hypothesis, additional sensors on the implants would be necessary to differentiate whether the femoral stem or the acetabular cup is loosened.

While the increase of signal energy transferred from the transducer to a loose implant and the decrease of signal energy transmitted across a loose bone-implant interface independently differentiated loose and fixed implants, the combination of both effects through the use of two sensors on both sides of the interface could be advantageous in more challenging clinical situations.

Special care has to be employed when only the signal energy on the bone side of the interface is measured, as there the two effects tend to cancel each other out. Such signal cancellation might also explain the results of another study on twelve porcine forelegs, which found no difference between press-fit and loosened implants when measuring the signal energy at the bone [34]. Their study also reported that central frequency decreased for once-replaced implants (33 N pull-out strength) compared to press-fit implants (90 N pull-out strength), and surprisingly dropped even more for twice-replaced implants (30 N pull-out strength) despite the minimal change of 3 N in pull-out strength. Interestingly, these findings might indicate that central frequency and signal energy reflect different interface properties.

While an exact micromotion value related to clinical implant stability is still under debate, stable implants are generally associated with interface micromotion of less than 150 μm [5,35–37]. While, to our knowledge, no pull-out strength values have been reported for press-fit shoulder implants, cemented hip implants exhibit pull-out strengths of about 2700 N for fixed and 230 N for loose implants [38]. The average micromotion (fixed: 70 μm, "loose": 125 μm)

**Table 1. Measurement uncertainties, statistical details, and differentiation performance for each acoustic characteristic, sensor location, and loading.** P-values were Bonferroni corrected with n = 35. Values for which the p-value exceeds 1.00 are not displayed.

| | Estimated uncertainty % | T-score, unloaded | p-value, unloaded | AUC, unloaded | T-score, loaded | p-value, loaded | AUC, loaded | T-score, LSR | p-value, LSR | AUC, LSR |
|---|---|---|---|---|---|---|---|---|---|---|
| **Mean total signal energy** | | | | | | | | | | |
| Implant sensor | 5.50 | -320.73 | < 0.001 | 1.00 | -190.33 | < 0.001 | 1.00 | 6.00 | 0.02 | 0.92 |
| Bone sensor | 16.79 | -27.47 | < 0.001 | 0.95 | -17.34 | < 0.001 | 0.86 | 1.34 | > 1.00 | - |
| Side sensor superior | 19.67 | -8.50 | < 0.001 | 0.77 | -44.66 | < 0.001 | 0.84 | -1.07 | > 1.00 | - |
| Side sensor inferior | 12.39 | -13.63 | < 0.001 | 0.73 | -67.15 | < 0.001 | 0.86 | -0.85 | > 1.00 | - |
| **Mean total signal energy ratio** | | | | | | | | | | |
| Implant sensor | - | - | - | - | - | - | - | - | - | |
| Bone sensor | 15.45 | 11.74 | < 0.001 | 1.00 | 10.72 | < 0.001 | 1.00 | -1.36 | > 1.00 | - |
| Side superior | 19.61 | 11.48 | < 0.001 | 0.70 | 2.02 | < 0.001 | 0.53 | -2.13 | > 1.00 | - |
| Side inferior | 13.86 | 15.47 | < 0.001 | 0.80 | 3.28 | < 0.001 | 0.53 | -1.96 | > 1.00 | - |
| **Harmonic ratio** | | | | | | | | | | |
| Implant sensor | 45.14 | -2.51 | 0.02 | 0.80 | -0.03 | 0.49 | 0.53 | - | - | |
| Bone sensor | 41.27 | 1.82 | 0.06 | 0.36 | 0.52 | 0.31 | 0.45 | - | - | |
| Side superior | 59.50 | 0.05 | 0.48 | 0.56 | -2.86 | 0.01 | 0.80 | - | - | |
| Side inferior | 34.86 | 1.12 | 0.15 | 0.48 | -4.16 | 0.002 | 0.75 | - | - | |

and pull-out strength (fixed: 760 N, loose: 271 N) measured during our tests show that our method was able to distinguish between comparatively small differences in fixation compared to these reported values. It seems furthermore plausible that with increasing looseness of the implant, more and more signal energy would be imparted to the implant-bone system while less and less signal energy would be transmitted across the interface, until no acoustic signal energy crosses the interface anymore. Such a progression would facilitate implant state quantification beyond the binary differentiation investigated in our study, and thus open perspectives for diverse monitoring applications. However, further studies involving a broader range of fixation states are clearly required to investigate the exact functional relationship between acoustic characteristics and implant stability.

The harmonic ratios in the implant sensor and bone sensor under loading, and in both side sensors under no loading, were able to differentiate between loose and stable implants with AUCs between 0.74 and 0.80 (Table 1). However, harmonic ratios showed high measurement uncertainties and varied considerably across sensors, making this acoustic characteristic a less promising candidate for clinical translation compared to MTSE or MTSER. Previous work has shown that harmonic ratios were able to distinguish between cemented implants, implants with a thin layer of silicone between bone and implant (mimicking fibrous tissue), and loose/fixed press-fit implants, but not between the latter two states [39]. The discrepancies between these results may originate from differences in creating the loose implants, as the previous work defined the implant states via the implantation procedure (a pull-out/replacement procedure) rather than through independent mechanical tests. Preliminary tests during development of our study showed that for our implant and test method, such a pull-out/replacement strategy did not create a loose condition as quantified by micromotion and pull-out strength. Even though this remains to be confirmed in further studies, together these results suggest that the previous work may have been unsuccessful in creating a loose implant condition and that harmonic ratios could potentially differentiate between all four implant states, including loose implants.

Considerable variations were observed for all acoustic characteristics between samples in loaded and unloaded conditions. Surprisingly, these variations did not appear consistent across specimens despite their homogeneity in terms of micromotion and pull-out strength. Furthermore, the assumption that the acoustic characteristics of stable implants change less under applied load than those of loose implants was shown to be false. These results could indicate that additional specimen properties that were not controlled in this study, such as implant tilt or implant-bone contact area, are involved in determining the value of each acoustic characteristic under a specific loading. While these effects were small enough to not deteriorate the differentiation between loose and fixed implants using absolute thresholds in most cases, they strongly affected LSR values, which were calculated using both loading conditions. The only LSR coefficient found to significantly differentiate between loose and fixed implant states was based on the MTSE in the implant sensor (AUC = 0.92, Fig 8). Because the differentiation performance based on MTSE was already perfect (AUC = 1.0), an improvement in performance due to adding LSR could not be measured and LSR remains unclear. While further investigations are needed, it is plausible that the additional information could prove valuable in more challenging clinical settings.

Limitations of our study prevent the direct transfer of our results to in-vivo measurements. As discussed above, the bone analog can only be considered a rough approximation of real bone. Even though the analog's general mechanical properties agree well with those of real bone, other properties such as anisotropy, porosity, as well as presence of fluid phases and soft-tissues are not represented in our model. Studies on real bone will thus be required to investigate whether our findings hold in the presence of these additional complexities. Further limitations include the use of a single implant/bone model, so the effects of implant design and bone anatomy remain to be studied. In this respect, a systematic study on the influence of implant type and design might allow the insights gained from studies on different implants and joints to be consolidated towards a more general, unified understanding. Here, modeling techniques may also provide valuable insights.

While further cadaveric and in-vivo studies are clearly required to confirm our findings, the fact that our quantitative acoustic approach was sensitive enough to detect the comparatively small differences in micromotion and pull-out strength, renders us optimistic for an eventual translation into clinical application. Where the limits in micromotion that still allow reliable differentiation lie, however, remains to be determined. For the intra-operative assessment of primary stability, the locations of the implant and bone sensors, which showed the best differentiation performance, are easily accessible during surgery, and loading for load self-referencing could be applied to the implant by the surgeon. As techniques to measure acoustic waves through the soft-tissues have shown promising results [40,41], our technique may also have the potential to be used for the non-invasive assessment of secondary stability during post-operative follow-up in which tranducer and sensor would be placed directly on the skin of the patient.

While achievement of primary stability is primary a concern for uncemented implants, such an assessment of secondary stability could potentially be useful for cemented implants as well. However, the presence of cement adds an additional layer of complexity—while only one interface (implant-bone) has to be measured for uncemented implants, for cemented implants the stabilities of two or even three interfaces (implant-cement, cement-bone, potentially implant-bone) need to be evaluated. We believe it likely that the technique could be applied to cemented implants, but further research and development will be needed.

In conclusion, quantitative acoustic measurements are able to differentiate between fixed and loose implants, reaching maximum AUC = 1.0 for mean total signal energy, AUC = 1.0 for mean total signal energy ratio, and AUC = 0.8 for harmonic ratio. Implant loosening

increased the signal energy transferred from the transducer to the implant but decreased the signal energy transmitted across the bone-implant interface. Even though load self-referencing could differentiate between implant states with AUC = 0.92, the benefits of combining load self-referencing with other acoustic characteristics remain to be shown. While the potential of this approach will need to be confirmed in clinical settings, cross-interface quantitative acoustics could ultimately facilitate the achievement of optimal primary stability during implantation surgery and avoid unnecessary implant revision through quantitative assessment of secondary stability during follow-up.

## Author Contributions

**Conceptualization:** Florian Vogl, Philippe Favre, William R. Taylor, Paul Thistlethwaite.

**Data curation:** Stefanie Greger, Philippe Favre, Paul Thistlethwaite.

**Formal analysis:** Florian Vogl, Stefanie Greger, Philippe Favre.

**Funding acquisition:** Philippe Favre, William R. Taylor, Paul Thistlethwaite.

**Investigation:** Florian Vogl.

**Methodology:** Florian Vogl, Philippe Favre, Paul Thistlethwaite.

**Project administration:** William R. Taylor, Paul Thistlethwaite.

**Resources:** William R. Taylor, Paul Thistlethwaite.

**Software:** Florian Vogl.

**Supervision:** Florian Vogl, Philippe Favre, William R. Taylor, Paul Thistlethwaite.

**Validation:** Florian Vogl, Paul Thistlethwaite.

**Visualization:** Florian Vogl, Stefanie Greger, Paul Thistlethwaite.

**Writing – original draft:** Florian Vogl, Paul Thistlethwaite.

**Writing – review & editing:** Florian Vogl, Paul Thistlethwaite.

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
