## [Decision Letter · Decision Letter 0]

13 Jan 2020

PONE-D-19-19911

Differentiation between mechanically loose and fixed press-fit implants using quantitative acoustics and load self-referencing: A phantom study

PLOS ONE

Dear Vogl,

Thank you for submitting your manuscript to PLOS ONE. After careful consideration, we feel that it has merit but does not fully meet PLOS ONE’s publication criteria as it currently stands. Therefore, we invite you to submit a revised version of the manuscript that addresses the points raised during the review process.

We would appreciate receiving your revised manuscript by Feb 27 2020 11:59PM. To enhance the reproducibility of your results, we recommend that if applicable you deposit your laboratory protocols in protocols.io, where a protocol can be assigned its own identifier (DOI) such that it can be cited independently in the future. For instructions see: http://journals.plos.org/plosone/s/submission-guidelines#loc-laboratory-protocols

We look forward to receiving your revised manuscript.

Kind regards,

David Fyhrie

Academic Editor

PLOS ONE

Journal Requirements:

3. We note that you have a patent relating to material pertinent to this article.

a. Please provide an amended statement of Competing Interests to declare this patent (with details including name and number), along with any other relevant declarations relating to employment, consultancy, patents, products in development or modified products etc. Please confirm that this does not alter your adherence to all PLOS ONE policies on sharing data and materials, as detailed online in our guide for authors http://journals.plos.org/plosone/s/competing-interests by including the following statement: "This does not alter our adherence to  PLOS ONE policies on sharing data and materials.” If there are restrictions on sharing of data and/or materials, please state these. Please note that we cannot proceed with consideration of your article until this information has been declared.

<h3>** **</h3>

4. Thank you for stating the following in the Financial Disclosure section:

'Unfunded studies'

We note that one or more of the authors are employed by a commercial company: Zimmer Biomet

Reviewers' comments:

Reviewer's Responses to Questions

**Comments to the Author**

1. Is the manuscript technically sound, and do the data support the conclusions?

Reviewer #1: Yes

2. Has the statistical analysis been performed appropriately and rigorously? 

Reviewer #1: Yes

3. Have the authors made all data underlying the findings in their manuscript fully available?

Reviewer #1: Yes

4. Is the manuscript presented in an intelligible fashion and written in standard English?

Reviewer #1: Yes

5. Review Comments to the Author

Reviewer #1: This paper aims to address a clinical problem (measuring implant stability) by using a non-invasive and radiation free technique. Using quantitative acoustics methods, the authors have compared the acoustic characteristics of a loose and fixed implant; showing the possibility of assessing the mechanical properties of the bone-implant interface and identifying between a loose and fixed implant.

A very well written paper. A few minor edits and comments below:

Line 100: "This implant was chosen because achievement of sufficient primary stability is central..." So, does this implant provide high stability?

Line 113-114, Figure 1 description: "a) labels the engulfing bone cement and b) the access slot for sensor mounting". What is this referring to? Does figure 1 require labelling?

Line 288: "Even though both effects,..." wording is awkward and may be missing a term.

Discussion: This study has been looking at loose and press-fit fixations. It may be good to comment (in the discussion) on how a cemented fixation would be different to current results.

6. PLOS authors have the option to publish the peer review history of their article (what does this mean?). If published, this will include your full peer review and any attached files.

Reviewer #1: No

---

## [Author Response · Author response to Decision Letter 0]

7 Feb 2020

Thank you for your valuable feedback, which has helped to improve our manuscript. Please kindly see the attached Response to Reviewers file for details.

---

## [Decision Letter · Decision Letter 1]

27 Apr 2020

PONE-D-19-19911R1

Differentiation between mechanically loose and fixed press-fit implants using quantitative acoustics and load self-referencing: A phantom study

PLOS ONE

Dear Vogl,

Thank you for submitting your manuscript to PLOS ONE. After careful consideration, we feel that it has merit but does not fully meet PLOS ONE’s publication criteria as it currently stands. Therefore, we invite you to submit a revised version of the manuscript that addresses the points raised during the review process

Your study is really  interesting. However, the title promises to much. Therefore I suggest to add some remarks, e.g. „A phantom study using polyurethane foam mimicking shoulder prosthesis“ or something like that. Furtermore, I recommend to add some Limitation in the abstract as well. From polyurtethane foam to clinical application is a long way to go. Nevertheless, you should try it.

We would appreciate receiving your revised manuscript by Jun 11 2020 11:59PM. To enhance the reproducibility of your results, we recommend that if applicable you deposit your laboratory protocols in protocols.io, where a protocol can be assigned its own identifier (DOI) such that it can be cited independently in the future. For instructions see: http://journals.plos.org/plosone/s/submission-guidelines#loc-laboratory-protocols

We look forward to receiving your revised manuscript.

Kind regards,

Hans-Peter Simmen, M.D., Professor of Surgery

Academic Editor

PLOS ONE

Reviewers' comments:

Reviewer's Responses to Questions

**Comments to the Author**

1. If the authors have adequately addressed your comments raised in a previous round of review and you feel that this manuscript is now acceptable for publication, you may indicate that here to bypass the “Comments to the Author” section, enter your conflict of interest statement in the “Confidential to Editor” section, and submit your "Accept" recommendation.

Reviewer #1: All comments have been addressed

Reviewer #2: All comments have been addressed

2. Is the manuscript technically sound, and do the data support the conclusions?

Reviewer #1: Yes

Reviewer #2: Yes

3. Has the statistical analysis been performed appropriately and rigorously? 

Reviewer #1: Yes

Reviewer #2: Yes

4. Have the authors made all data underlying the findings in their manuscript fully available?

Reviewer #1: Yes

Reviewer #2: Yes

5. Is the manuscript presented in an intelligible fashion and written in standard English?

Reviewer #1: Yes

Reviewer #2: Yes

6. Review Comments to the Author

Reviewer #1: The authors have adequately addressed all comments and queries.

Reviewer #2: This biomechanical study investigates the mechanical stability of an shoulder implant without stem an sawbone model of polyurethane foam. The study shows clearly that a differentiation between loose and stable implants is possible in this biomechanical model by acoustic measurements. The methods and quantitative acoustic measurements are described meticulously.

Loosening of prosthetic replacement is more likely a problem at the lower extremities in hip and knee replacement. Because of this secondary loosening of this replaced joints is related to mechanical load in obese patients (found among others reasons: aseptic/septic loosening).

A shoulder prosthesis in an “unloaded prosthesis”. Early loosening after implantation of a shoulder prosthesis is rare and more related to other reasons: intraoperative fracture/fissure, infection, osteoporosis).

The evaluation of sufficient primary stability (called press-fit) of an implantat during the operation is difficult. This is clinical experience of the orthopedic surgeon. Proper preoperative planning of the implants at the template and correct reaming of the bone helps to achieve adequate primary implant stability in the bone.

From clinical prospective good primary stability and press-fit of the implant is important. But various independent factors influence stable osseointegration of the implant. Furthermore, a customized postoperative treatment regimen is necessary avoiding excessive micromotion of the implant in the first week after surgery.

“59 and thus have limited applicability for inter-operative assessment.”?

As mentioned in the paper evaluation of secondary loosening of implants remains hard to detect. In practice most suitable diagnostic approaches are CT scan and if available SPECT-CT. The examinations should, respectively must, clarify implant loosening before an intervention. This has to be discussed with the patient. CT scan and SPECT-CT are non-invasive.

Acoustic techniques require direct contact of piezo-electric actuator and the accelerometer with bone and implant. An intraoperative (invasive) testing is possible. However loose implants are easy to identify by manual testing during the intervention in the operating theatre. The authors should clarify how acoustic techniques can offer simple and non-invasive assessment of prosthetic loosening in approximation to a clinical case to avoid confusion with the biomechanical model and non-invasive techniques.

A critical point bringing these techniques into clinical practice seems to be the specificity of the acoustic measurements on the implant type, joint und anatomy, which makes interpretation of the measurements difficult.

The new approach “load self-referencing (LSR)” with application of a load to the implant changing the acoustic properties of an unstable implant would be a promising further development. This approach would serve as their own reference. But the acoustic characteristics under different loading conditions were not plausible. Further investigations are needed.

Furthermore, translation in clinical practice in the operating theatre seems to be difficult. A loaded versus unloaded position/condition of an extremity (single-legged stance for hip and knee implant?) in the operating theatre is not possible also in respect to aseptic technique and the surrounding soft tissue.

7. PLOS authors have the option to publish the peer review history of their article (what does this mean?). If published, this will include your full peer review and any attached files.

Reviewer #1: No

Reviewer #2: No

---

## [Author Response · Author response to Decision Letter 1]

7 May 2020

We thank the Reviewers for their feedback and helping us improve our manuscript. Please see the dedicated Reponse to Reviewers file for details.

---

## [Editor Report · Decision Letter 2]

8 May 2020

Differentiation between mechanically loose and fixed press-fit implants using quantitative acoustics and load self-referencing: A phantom study on shoulder prostheses in polyurethane foam

PONE-D-19-19911R2

Dear Dr. Vogl,

We are pleased to inform you that your manuscript has been judged scientifically suitable for publication and will be formally accepted for publication once it complies with all outstanding technical requirements.

With kind regards,

Hans-Peter Simmen, M.D., Professor of Surgery

Academic Editor

PLOS ONE
---

## [Editor Report · Acceptance letter]

19 May 2020

PONE-D-19-19911R2 

Differentiation between mechanically loose and fixed press-fit implants using quantitative acoustics and load self-referencing: A phantom study on shoulder prostheses in polyurethane foam 

Dear Dr. Vogl:

I am pleased to inform you that your manuscript has been deemed suitable for publication in PLOS ONE. Congratulations! Your manuscript is now with our production department. 

With kind regards,

on behalf of

Dr. Hans-Peter Simmen 

Academic Editor

PLOS ONE